# A Newborn Screening Program for Sickle Cell Disease in Murcia (Spain)

**DOI:** 10.3390/ijns9040055

**Published:** 2023-10-10

**Authors:** María Sánchez-Villalobos, Eulalia Campos Baños, María Jesús Juan Fita, José María Egea Mellado, Inmaculada Gonzalez Gallego, Asunción Beltrán Videla, Mercedes Berenguer Piqueras, Mar Bermúdez Cortés, José María Moraleda Jiménez, Encarna Guillen Navarro, Eduardo Salido Fierrez, Ana B. Pérez-Oliva

**Affiliations:** 1Hematology Service, Virgen de la Arrixaca University Hospital, 30120 Murcia, Spain; maria29sv@gmail.com (M.S.-V.); jmoraled@um.es (J.M.M.J.); 2Biomedical Research Institute of Murcia (IMIB), 30003 Murcia, Spain; lalycampos19@gmail.com; 3Biochemistry and Clinical Genetics Center, Virgen de la Arrixaca University Hospital, 30120 Murcia, Spain; mariaj.juan2@carm.es (M.J.J.F.); josem.egea2@carm.es (J.M.E.M.); inma.gonzalez@carm.es (I.G.G.); 4Pathological Anatomy Service, Santa Lucia University Hospital, 30202 Cartagena, Spain; asuncionbeltranvi@gmail.com; 5Hematology Service, Morales Meseguer Hospital, 30008 Murcia, Spain; mers79@gmail.com; 6Pediatric Service, Virgen de la Arrixaca University Hospital, 30120 Murcia, Spain; mariam.bermudez2@carm.es (M.B.C.); encarga.guillen@carm.es (E.G.N.)

**Keywords:** sickle cell disease, vaso-occlusive crises, structural hemoglobinopathies, newborn screening

## Abstract

Sickle cell disease (SCD) is an inherited autosomal recessive hemoglobin disorder caused by the presence of hemoglobin S, a mutant abnormal hemoglobin caused by a nucleotide change in codon 6 of the β-globin chain gene. SCD involves a chronic inflammatory state, exacerbated during vaso-occlusive crises, which leads to end-organ damage that occurs throughout the lifespan. SCD is associated with premature mortality in the first years of life. The process of sickling provokes asplenia in the first years of life with an increased risk of infection by encapsulated germs. These complications can be life-threatening and require early diagnosis and management. The most important interventions recommend an early diagnosis of SCD to ensure that affected newborns receive immediate care to reduce mortality and morbidity. The newborn screening program in the region of Murcia for SCD began in March 2016. We aimed to determine the incidence of sickle cell anemia and other structural hemoglobinopathies in the neonatal population of the region of Murcia, an area of high migratory stress, and to systematically assess the benefit of newborn screening for SCD, leading to earlier treatment, as well as to offer genetic counseling to all carriers. The prevalence of SCD in our region is similar to others in Spain, except for Catalonia and Madrid. The newborns with confirmed diagnoses of SCD received early attention, and all the carriers received genetic counseling.

## 1. Introduction

Sickle cell disease (SCD) is an inherited autosomal recessive hemoglobin (Hb) disorder caused by the presence of hemoglobin S, a mutant abnormal hemoglobin caused by a nucleotide change in codon 6 of the β-globin chain gene, which leads to the coding of the amino acid valine instead of glutamic acid in the sixth position of the β-globin chain [1].

The presence of severe forms of SCD include the homozygous state (SS) or the co-inheritance of HbS with the β0 thalassemia mutation. Other less severe forms include the co-inheritance of HbS with other β-globin gene mutations such as hemoglobin C, hemoglobin D, or β+ thalassemia mutation [2].

Hemoglobin S is characterized by reduced solubility and increased polymerization (sickling) under hypoxic conditions, leading to the formation of sickled red blood cells that change their deformability and adhesiveness to the vascular endothelium. Moreover, SCD involves a chronic inflammatory state, exacerbated during vaso-occlusive crises, which leads to end-organ damage that occurs throughout the lifespan [1,3].

SCD is associated with premature mortality with a median age of death of 43 years (IQR 31.5–55 years) [4], but especially during the first years of life, where patients with SCD are at high risk of life-threatening complications. The process of sickling provokes asplenia in the first years of life with an increased risk of infection via encapsulated germs. These complications can be life-threatening, difficult to detect, and require early diagnosis and management [1,3].

Infections are known to be the main cause of mortality in children with SCD, with *Streptococcus pneumoniae* as the main etiological agent. Early initiation of antibiotic prophylaxis with oral penicillin was shown to reduce the rate of *Streptococcus pneumoniae* infection by 84% [3] and has therefore become a standard treatment in the European and international guidelines for children aged 0–5 years [5,6].

In 1986, a multicenter randomized, placebo-controlled study demonstrated the benefit of using prophylactic penicillin in children with SCA (HbSS) between the ages of 3 and 36 months old. This study showed an 84% reduction in the incidence of septicemia from Streptococcus pneumoniae observed in the treatment group [7]. This study supported the importance of an SCD NBS program to ensure early intervention. This allows for early diagnosis and a comprehensive approach that includes family education to detect symptoms of alarm and infectious prophylaxis. Other studies confirmed the benefit of mortality reduction (Vinchisky et al., 1998 and the Jamaican Cohort Study of SCD, 1995) [8,9].

It is essential to recognize early severe complications that may be life-threatening, such as abdominal sequestration crisis. Parents should receive the necessary training to recognize the signs of these complications and to treat them as medical emergencies.

Accordingly, the most important recommendations, such as the World Health Organization [10], the US guide on newborn screening [11], and the British National Health Service [12], recommend an early diagnosis of SCD to ensure that affected newborns receive immediate care to reduce mortality and morbidity.

In Spain, screening for SCD was incorporated into the National Health System’s service portfolio in November 2014, and all the Regional Health Services have a neonatal screening program for sickle cell anemia. In Murcia, the screening program began in March 2016.

## 2. Objective

We aimed to determine the incidence of sickle cell anemia and other structural hemoglobinopathies in the neonatal population of the region of Murcia, an area of high migratory stress, and to systematically assess the benefit of newborn screening for SCD leading to earlier treatment.

One of the dilemmas that arises is whether this screening should be universal or selective, that is, directed only at newborns from areas with a high prevalence of these diseases. Therefore, we also analyze the incidence among the immigrant population according to the native population and the usefulness of genetic counseling offered to families of carriers.

## 3. Methodology

### 3.1. Subjects

The newborn program in the region of Murcia includes the detection of SCD in any of its variants: homozygosis or double heterozygosis in combination with other hemoglobinopathies (C, D, E, β^0^-thalassemia, and β^+^-thalassemia) for all newborns babies in the region of Murcia. In Spain, SCH NBS is universal, and all newborns participated in the study.

### 3.2. Hemoglobin Variant Analysis

Samples obtained from heel blood were collected at hospital discharge (at 48 h of life), excluding samples from transfused patients in whom the study was postponed up to three months of life.

The technique used in the initial screening was capillary electrophoresis using the Capillary Neonatal kit supplied by Sebia^®^. For detection, 3.6 mm diameter samples of blood impregnated in the paper were perforated using the Panthera Puncher^TM^ 9 equipment from the commercial company Perkin Elmer^®^ and deposited into 8-well segments. The addition of 50 µL of ultrapure water to each well of each segment and subsequent incubation for 20–24 h in a humidified chamber at 2–8 °C allowed hemolysis of the red blood cells, whose hemoglobin can be directly analyzed using the Capillary Electrophoresis System—Sebia 2 NeonatFastTM. The barcode numbering of the samples allowed complete traceability of the results obtained.

After the detection of a structural hemoglobin variant in the screening, a referral was made to the Hematology Department of the Hospital Clínico Universitario Virgen de la Arrixaca (HCUVA), where an attempt was made to confirm the results obtained in the newborn, extend the study to the progenitors, and provide genetic counseling in all cases. After obtaining informed consent, a sample of peripheral blood anticoagulated with EDTA was collected by venipuncture and first subjected to capillary electrophoresis using the MINICAP system (Sebia^®^), which automatically performed all the stages of electrophoresis.

The system reads the barcodes on the sample tubes and performs sample dilution using disposable reagent domes, including an anode reservoir. It used two capillary tubes and cleansed them using different solutions (MINICAP washing solution, distilled water, and/or analysis buffer). In each column, an internal quality control was conducted. The samples were then injected into the capillaries by bringing one end of the capillaries into contact with the diluted samples, aspirating a very small volume of sample into each capillary. It then performed migration at constant temperature using a Peltier effect system and detected the separated fractions by absorbance spectrophotometry. Finally, the results were edited using PHORESIS software (Sebia^®^, Lisses, France).

Following the detection of a hemoglobin variant, the mutational study of the ß-globin gene was performed using the β-Globin StripAssay ^®^ method (ViennaLab, Vienna, Austria), based on polymerase chain reaction (PCR) and reverse-hybridization. This procedure included DNA isolation, PCR amplification using biotinylated primers, and the hybridization of amplification products to a test strip containing allele-specific oligonucleotide probes immobilized as an array of parallel lines. The assay covered the 22 most frequent mutations of the β-globin gene in the Mediterranean area, namely -101 [C>T], -87 [C>G], -30 [T>A], codon 5 [-CT], codon 6 [G>A] HbC, codon 6 [A>T] HbS, codon 6 [-A], codon 8 [-AA], codon 8/9 [+G], codon 15 [TGG>TGA], codon 27 [G>T] Knossos, IVS 1.1 [G>A], IVS 1.5 [G>C], IVS 1.6 [T>C], IVS 1.110 [G>A], IVS 1.116 [T>G], IVS 1.130 [G>C], codon 39 [C>T], codon 44 [-C], IVS 2.1 [G>A], IVS 2.745 [C>G], IVS 2.848 [C>A], and > 90% of β-globin defects found in Mediterranean countries.

The same technique described for the β-chain study was used for the α-thalassemia cases detected, including in this case the 21 most frequent mutations of the α-globin gene, namely 3.7 single gene deletion, 4.2 single gene deletion, MED double gene deletion, SEA double gene deletion, THAI double gene deletion, FIL double gene deletion, 20.5 kb double gene deletion, anti-3.7 gene triplication, α1 cd 14 [TGG>TAG], α1 cd 59 [GGC>GAC] (Hb Adana), α2 init cd [ATG>ACG], α2 cd 19 [-G], α2 IVS1 [-5nt], α2 cd 59 [GGC>GAC], α2 cd 125 [CTG>CCG] (Hb Quong Sze), α2 cd 142 [TAA>CAA] (Hb Constant Spring), α2 cd 142 [TAA>AAA] (Hb Icaria), α2 cd 142 [TAA>TAT] (Hb Pakse), α2 cd 142 [TAA>TCA] (Hb Koya Dora), α2 poly A-1 [AATAAA-AATAAG], α2 poly A-2 [AATAAA-AATGAA] (Hb Koya Dora), α2 poly A-1 [AATAAA-AATAAG], and α2 poly A-2 [AATAAA-AATGAA] (Hb Koya Dora).

In some selected cases, complete sequencing of the beta-globin gene was carried out.

## 4. Results

The results are based on a single retrospective study. From March 2016 to March 2023, a total of 104,083 newborns from all the hospitals of the Murcia region were included in the study. These babies were referred from different health areas to the Hematology Department of the HCUVA for detection of an abnormal hemoglobin variant in neonatal screening (Figure 1).

The incidence per 100,000 inhabitants in the region is 3.02. The incidence per 100,000 inhabitants among the different health areas is similar. The health area with the highest incidence was area III, probably due to the greater migratory flow (Table 1) [13].

The total number of cases detected with some hemoglobin variants was 498 (4.8 of 1000 cases). Of these, five (0.05 of 1000 cases) were classified as homozygous SS (SCA) with neonatal FS phenotype. Other sickle cell syndromes were detected in combination with other haemoglobinopathies (hemoglobin C, FSC phenotype) in five (0.05 of 1000 cases). Therefore, the global sickle cell syndromes detected in our study were 10 cases (0.1 of 1000 cases).

The remaining hemoglobin variants (no sickle cell syndromes) detected corresponded to 285 carriers (2.7 of 1000 cases) for hemoglobin S (sickle cell trait), with neonatal FAS phenotype, 182 cases for carriers for hemoglobin C (FAC phenotype) and 17 cases were carriers for others hemoglobin’s variants (G-Ph hemoglobin, D- hemoglobin, or double heterozygosis hemoglobin (C-Korle Bu double hemoglobin).

It is interesting to highlight that four cases were homozygous CC, a condition that can be associated with hemolytic anemia. Finally, three cases of screening failure were detected (0.6%), in which the presence of any variant was not confirmed (Table 2).

Regarding the hemoglobin variant detected, 2.8% of the newborns were homozygous or double heterozygous for β-chain structural variants (SS, SC, and CC), confirming heterozygosity in the bulk of the newborns analyzed (Table 2). Of the latter group, in 4.8% of the cases studied, both parents were double heterozygous: 3.4% AS/AS, 0.2% AS/AC, 0.2% AS/O-Arabia, and 1% AS/α-thalassemia.

Of the total newborns studied, only 3.6% of both parents were of European origin, and in the rest of the cases studied, at least one of the parents was of African or South American origin (Hispanic). Out of 943 parents analyzed, 70% were from the African continent, although the country of origin was only collected in 700 of those studied (Figure 2).

Of the 943 parents studied, 1.7% were homozygous or double heterozygous for the β-chain structural variant: 0.1% SS, 0.6% SC, and 1% CC (Table 3). Although not a target for neonatal screening, 1.2% of silent α-thalassemia carriers were detected. Of this group, 27.3% had offspring with α-thalassemia, 27.3% were not detected, and 45.4% of newborns were not screened due to insufficient samples.

## 5. Discussion

This is the first time that data on neonatal screening in the region of Murcia have been analyzed since its implementation in March 2016. The prevalence of SCD in our region is approximately 1 case per 10,000 newborns, lower than the data reported by communities such as Catalonia (1/3155), Baleares (1/6756), and Madrid (1/5552). This is probably because, in our region, the migratory flow is lower [14]. Except for Catalonia and Madrid, the regions with the highest migratory flow, our data agree with what is reported in other regions of Spain, such as Western Andalusia (1/12356) [14,15].

In our region, neonatal screening for SCD was implemented within the preventive medicine measures included in public health programs, which should lead to early medical intervention for the benefit of the newborn (Committee on Screening for Inborn Errors of Metabolism, Genetic Screening) [16].

Neonatal screening is used to diagnose most newly diagnosed patients in our region, with only exceptional cases diagnosed in adults. Since its implementation in 2016, neonatal screening has had two main objectives: early detection of SCD cases for referral to the Pediatric Hematology Clinic to initiate antibiotic prophylaxis with oral penicillin; and detection of heterozygous cases to offer genetic counseling to reduce the number of affected cases at birth. In 3.8% of parents offered genetic counseling, both parents were heterozygous for the β-chain structural variant (3.4% AS/AS, 0.2% AS/AC, and 0.2% AS/O-Arabia), at risk of having a child with sickle cell disease.

Genetic counseling is difficult to implement in daily practice due to language barriers and socio-cultural and work-related problems inherent to the more susceptible population. These issues result in incomplete family studies (6%), absence at medical appointments, and difficulty in understanding genetic counseling.

Taking into account that the mortality in children under 5 years of age is high at 30%, and despite the fact that the incidence is not as high as in other regions and that in all cases detected, the parents were of non-European origin, we believe that the SCD screening should be universal, as recommended by ANHI [17]. However, some groups consider selective screening to be more effective [18,19].

In Spain, there is little information on the true incidence of the different forms of hemoglobinopathies. Our data can contribute to a better knowledge of the incidence of sickle cell disease in the different Mediterranean regions and encourage extending screening to those regions where it is not implemented and performing genetic counseling in all detected carriers of sickle cell trait.

Due to increasing migratory movements, neonatal screening programs in Western countries have become indispensable. Early diagnosis of SCD allows for optimal treatment with a decrease in infant morbidity and mortality. All the newborns detected in the program have benefited from early medical attention, continue to be monitored in our pediatric unit, and have not developed any complications during the follow-up of the study.

Despite its difficulty, there is a need to continue to improve the communication skills with parents, and offer genetic counseling to all detected carriers in a simple way and in a language they can understand, and always confirm that the message conveyed has been understood.

## 6. Limitations and Strengths

Due to the retrospective nature of the study, in some cases, not all data could be collected. On limited occasions, we were not able to have access to both parents or they have not come to the appointment. This is due to the social situation of this population, mostly immigrants and with few resources. However, the study has strengths. It is the first study carried out in the region of Murcia that offers us an estimate of the prevalence of SCD and other hemoglobinopathies in a population. The fact that we are a small autonomous community has allowed us to implement not only universal screening but universal genetic counseling for all carriers (sickle cell trait). All the newborns detected in the program have benefited from early medical attention, continue to be monitored in our pediatric unit, and have not developed any complications during the follow-up of the study.

## Figures and Tables

**Figure 1 IJNS-09-00055-f001:**
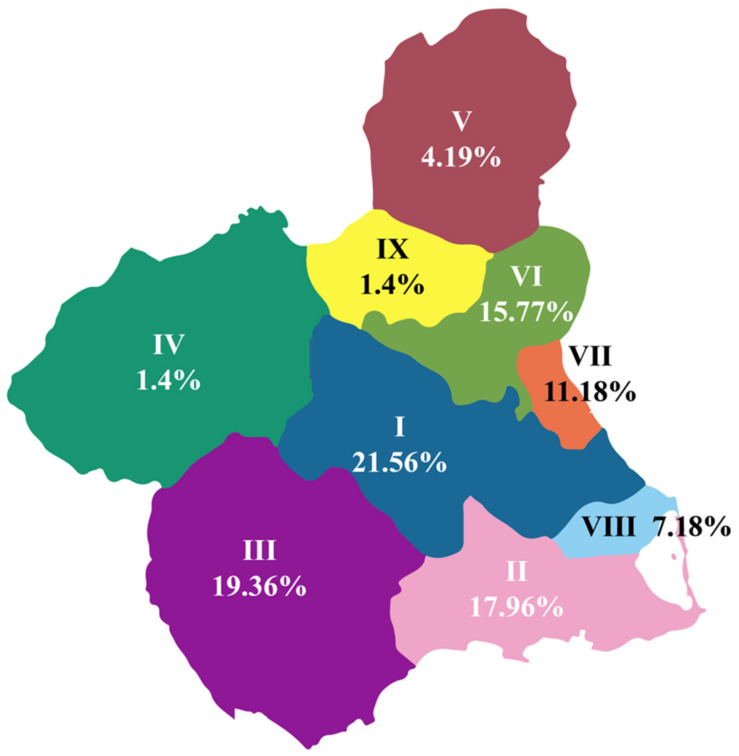
Proportion of patients (%) referred from the nine health areas that belong to the Murcian health system.

**Figure 2 IJNS-09-00055-f002:**
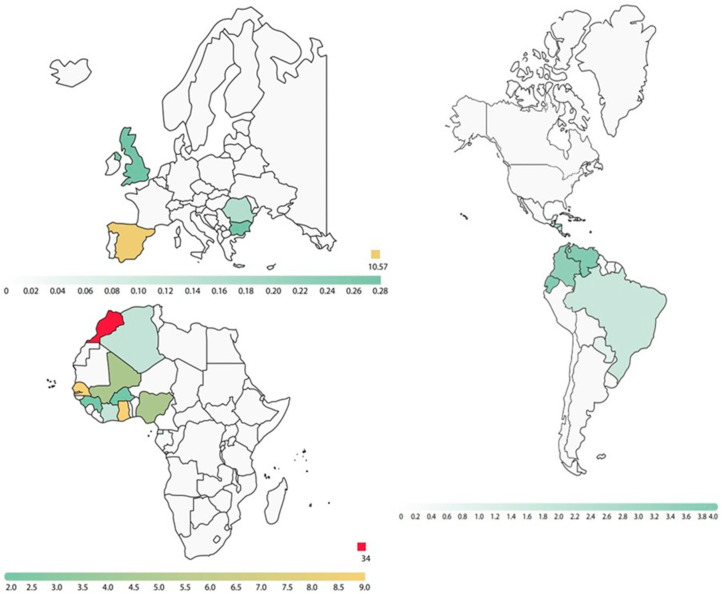
Color intensity graph showing the frequency (%) of the countries of origin of the parents (700 known cases out of a total of 943 studied).

**Table 1 IJNS-09-00055-t001:** Incidence of abnormal hemoglobin variant per health area.

Health Area	%	Total Cases	Population [13]	Incidence (per 100,000)
I	21.56	108.0156	269,627	4.01
II	17.96	89.9796	288,536	3.12
III	19.36	96.9936	180,577	5.37
IV	1.4	7.014	69,947	1.00
V	4.19	20.9919	60,828	3.45
VI	15.77	79.0077	272,042	2.90
VII	11.18	56.0118	204,969	2.73
VIII	7.18	36.0219	109,851	3.28
IX	1.4	7.014	54,874	1.28
Total	100	501	1,511,251	
Average Incidence				3.02

**Table 2 IJNS-09-00055-t002:** Hemoglobin variants detected in newborns in neonatal screening.

Hemoglobin Variant	Number (%)
Sickle Cell Syndromes	
SS	5 (1)
SC	5 (1)
Carriers (heterozygous)	
Hemoglobin S	285 (57)
Hemoglobin C	182 (36.5)
Hemoglobin G-Philadelphia	7 (1.4)
Hemoglobin D	4 (0.8)
Hemoglobin E	1(0.2)
Hemoglobin J	1 (0.2)
Hemoglobin O-Arabia	1 (0.2)
Others	
CC	4 (0.8)
Alpha-thalassemia	2 (0.4)
Hemoglobin C/Korle-Bu	1 (0.2)

SS: hemoglobinopathy S in homozygosity; CC: hemoglobinopathy C in homozygosity SC: double heterozygosity for hemoglobin S and C.

**Table 3 IJNS-09-00055-t003:** Frequency of variants detected in parents according to country of origin.

Regions		Structural Variants Detected in Parents					
		AS	AC	SS	CC	SC	OTHER
**Africa**	**Northern**	57 (6%)	67 (7.1%)	-	2 (0.2%)	1 (0.1%)	2 (0.2%)
	**Occidental**	102 (10.8%)	49 (5.2%)	-	6 (0.6%)	4 (0.4%)	-
	**Central**	4 (0.4%)	-	-	-	-	-
	**Oriental**	-	-	-	-	-	-
	**Southern**	-	1 (0.11%)	-	-	-	-
	**Unknown**	32 (3.4%)	22 (2.3%)	-	2 (0.2%)	1 (0.1%)	2 (0.2%)
**America**		57 (6%)	20 (2.1%)	1 (0.1%)	-	-	1 (0.1%)
**Canada**		1 (0.1%)	-	-	-	-	-
**Europe**		9 (1%)	4 (0.4%)	-	-	-	6 (0.6%)

AS: haemoglobinopathy S in heterozygosity; AC: haemoglobinopathy C in heterozygosity; SS: haemoglobinopathy S in homozygosity; CC: haemoglobinopathy C in homozygosity SC: double heterozygosity for hemoglobin S and C; Other variants in heterozygosity: hemoglobin D, E, J, O-Arabia, G-Philadelphia, and Korle Bu.

## Data Availability

The data that support the findings of this study are available on request from the corresponding author. The data are not publicly available due to restrictions because their containing information that could compromise the privacy of research participants.

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
