# Peer review of "A Newborn Screening Program for Sickle Cell Disease in Murcia (Spain)"

_2409-515X, 2023, doi:10.3390/ijns9040055_

Round 1
Reviewer 1 Report
i have indicated some suggestion inthe text see attached
it is important that heterozygous states with co0existence of HbS are removed from the definition of SCD. I have also suggested some improvement in the text - in-approriate use of guidelines etc see attached

english can be improved and the paragraph layout to be done in line with the themes
Author Response
Thank you very much for taking the time to review this manuscript. Please find attached detailed responses below and the corresponding revisions/corrections changes in the re-submitted files.
“I have indicated some suggestion in the text see attached”
“It is important that heterozygous states with coexistence of HbS are removed from the definition of SCD. I have also suggested some improvement in the text - in-appropriate use of guidelines etc see attached”.
We agree with the comments in the text. I have modified them according to the suggestions. See attached in the text.
We agree with need to emphasize and separate cases of sickle cell syndromes from the rest, but we believe it is important to highlight the rest of the variant haemoglobin’s detected, which are the majority, and the importance of correct diagnostics, because they are not SCD. Check modified table 2 and explicative text on page 4.

Reviewer 2 Report
Please summarize the main findings and conclusion of the study in the abstract.
The authors report the success and challenges of new born screening programme in the hospitals of Murcia region, Spain. Please highlights limitations and strengths of the study.
Outcome of the second objective "systematically assess the benefit of NBS for SCD leading to earlier treatment" should be addressed properly.
The authors need to provide some rational for how they selected the infants to be referred to Hematology department from different health areas.
Authors should mentioned whether techniques used for the mutational analysis has been developed in house by them or is it a ready commercial kit ?
The study conducted over a period of 7 years. Any attempt has been made for clinical follow up of the SCD patients ?
There are useful information in this paper but revision is needed to improve the clarity.
Check the spellings in the text.
Author Response
Thank you very much for taking the time to review this manuscript. Please find attached detailed responses below and the corresponding revisions/corrections changes in the re-submitted files.
Reviewer 2
Please summarize the main findings and conclusion of the study in the abstract.
As well as offer genetic counseling to all carriers. The prevalence of SCD in our region is similar that others in Spain, except for Catalonia and Madrid. The newborns with confirmed diagnosis of SCD received early attention and all the carriers received genetic counseling.
The authors report the success and challenges of newborn screening programme in the hospitals of Murcia region, Spain. Please highlights limitations and strengths of the study.
Due to the retrospective nature of the study, in some cases not all data could be collected. On limited occasions we have not been able to have access to both parents or they have not come to the appointment. However, the study has strengths. It is the first study carried out in the Region of Murcia that offers us an estimate of the prevalence of SCD and other hemoglobinopathies in a population. The fact of being a small autonomous community has allowed us to implement not only universal screening, but universal genetic counselling for all carriers (sickle cell trait). All the newborn detected in the program have benefited from early medical attention, and continue to be monitored in our pediatric unit, and have not developed any complications during the follow-up of the study.
Outcome of the second objective "systematically assess the benefit of NBS for SCD leading to earlier treatment" should be addressed properly.
In 1986, a multicenter randomized, placebo-controlled study demonstrated the benefit of using prophylactic penicillin in children with SCA (HbSS) between the ages of 3 and 36 months old. These study showed an 84% reduction in the incidence of septicemia from Streptococcus pneumoniae observed in the treatment group (7). This study supported the importance of an SCD NBS program to ensure early intervention. Thus allows early diagnosis and a comprehensive approach that includes family education to detect symptoms of alarm and infectious prophylaxis. Othres studies confirmed the benefoct of mortality reduction (Vinchisky et al 1998 and the Jamaican Cohort Study of SCD, 1995) (8, 9).
The authors need to provide some rational for how they selected the infants to be referred to Hematology department from different health areas.
In Spain, SCH NBS is universal and all newborn participated in the study.
Authors should mentioned whether techniques used for the mutational analysis has been developed in house by them or is it a ready commercial kit?
Following the detection of a hemoglobin variant, the mutational study of the ß-globin gene was performed using the β-Globin StripAssay ® method (ViennaLab)
The study conducted over a period of 7 years. Any attempt has been made for clinical follow up of the SCD patients?
All the newborn detected in the program have benefited from early medical attention, and continue to be monitored in our pediatric unit, and have not developed any complications during the follow-up of the study.
